

# Climatic Feedbacks and Vegetation Changes Driven by Orbital Forcing and Sea Surface Temperature During Interglacials

Carlos Gurjão[1], Flávio Justino[1], Marcos Pereira[2], and Mônica Senna[3]

[1]Department of Agricultural Engineering, Federal University of Viçosa, Viçosa-MG, Brazil
[2]Department of Meteorology, Federal University Alagoas, Maceió-AL, Brazil
[3]Biosystems Engineering, Federal University Fluminense, Niterói-RJ, Brazil

**Correspondence:** Carlos Gurjão (carlosdiegogurjao@gmail.com)

**Abstract.** Climatic feedbacks associated with orbitally driven Sea Surface Temperature (SST) lead to a profound impact on the Southern Hemisphere (SH) vegetation cover during the Mid-Holocene, MIS5e, MIS11c, and MIS31 interglacials. Results are based on a suite of coupled climate simulations conducted with the ICTP-CGCM (Speed-Nemo), which provides the boundary conditions for the CCM3/IBIS vegetation model. The CCM3/IBIS model was run from the Speed-Nemo output of the global ocean and individualized Atlantic, Pacific and Indian ocean basins forcing, in addition to orbital parameters and greenhouse
gases. For interglacials, MH, MIS 5e, and 11c, areas have been found to be significantly reduced in tropical evergreen forests, but more extensive savanna and grasslands, over parts of the southern and central African region and into northern South America (Amazon region). In another important period, the results showed that there have been changes in vegetation cover due to insolation forcing during MIS31 compared to other interglacials, but the impact was greater in Australia and central South
America. However, the southern tropical climate became drier due to negative SST anomalies, induced by the Atlantic and Tropical Pacific basins, and thus reduced continental precipitation. This study demonstrates that vegetation responses across tropical regions of the Southern Hemisphere are not solely driven by orbital variations but are also significantly modulated by internal changes in Atlantic and Pacific SST patterns. In the study regions, in particular, during the MH, MIS5e and MIS11c, the seasonality of insolation was reduced in the SH, leading to the cooling of the southern tropical ocean basins, resulting in
the migration of the summer precipitation zone to boreal latitudes. Therefore, combined with increased low-latitude summer temperatures and a prolonged dryer period, the forcing led to a slow retraction, but steady in the rainforests of Congo and Amazonia, causing an effect of extreme aridity.

## 1   Introduction

During the last Pleistocene, vegetation cover experienced significant changes driven by climate variability, including orbital
forcing, fluctuations in sea surface temperature (SST), and greenhouse gas (GHG) concentrations (Lee et al., 2021; De Boer et al., 2021; Ganopolski and Calov, 2011; Lüthi et al., 2008). Previous studies using global circulation models (GCM) have already indicated that vegetation-atmosphere feedbacks in response to orbital parameters were responsible for climate change during the last Pleistocene and Holocene (Foley et al., 2000; Claussen, 1994; Foley et al., 1994). Several approaches, including paleoreconstructions, have explored changes in global vegetation of past interglacial, such as during the MIS 5e and MH, which



occurred about 6000, 120000 years before the present. However, the MIS 11c and 31 stages, which took place 411000 years ago, and 1.08M have not been investigated in detail on a global perspective based on global climate simulations (Shugart and Woodward, 2011). The ocean plays an important role in vegetation-atmosphere interactions, as the distributions of rainfall and tropical vegetation cover are strongly influenced by the variability of SST (Kim et al., 2021). In fact, oceanic changes on a geological time scale, mediated by interglacial periods, favor ecosystem changes worldwide, and because of modifications

in climate feedbacks (Vázquez-Rivera and Currie, 2015) can shed light on Earth's climate, which is currently experiencing remarkable changes in its thermal structure.

Changes in vegetation dynamics and climate variability have already been discussed by global political and scientific authorities in recent years (IPCC, 2013, 2014; Callaghan et al., 2020), once the impacts caused by climate change made the ecosystem more vulnerable (Brandt et al., 2017; Gao et al., 2018), the shrink of tropical vegetation leads to the reduction of

evapotranspiration, resulting in the destabilization of the hydrological cycle. Despite the direct existence of the relationship between the main parameters of climate (temperature and precipitation) and vegetation cover (Wu et al., 2016), investigating the responses of vegetation dynamics to climate change associated with the past offers us the opportunity to understand the distribution of landscape dynamics (Jamieson et al., 2012). According to a recent study, it indicated that photosynthetic activity will continue to decrease under conditions of increasing temperature and water deficit, especially in regions with climates

vulnerable to drought (Adepoju et al., 2019). Consequently, the impact of climate on populations that live in a situation vulnerable to poverty and that depend on subsistence agriculture, especially rural populations, will be much greater because there are still no strategic adaptation plans for future extreme events (Olsson et al., 2014).

Based on several climate modeling experiments, previous studies have reported past climate variations involving vegetation-precipitation during the Middle Holocene, in which precession-induced insolation dominated that epoch, causing SST elevation

in the global tropical region, northward migration from African summer monsoons and increased undergrowth in Sahara and Sahel regions (Collins et al., 2013; Kro¨pelin et al., 2008; Zhao et al., 2005), as well as the strengthening of the South American monsoon (Prado et al., 2013).

Paleovegetation studies conclude that climate changes and increase in $CO_2$ concentrations were responsible for changes in vegetation dynamics and composition around the world (Nolan et al., 2018; Prentice et al., 2011). However, the oceanic

component mostly governs energy storage in the climate system due to its high specific heat value and physical-chemical properties. Through coupled modeling, Braconnot et al. (2007) showed that African monsoon rain during the MH ($\approx$ 6 kyr ago) depends on ocean feedbacks, ie a strong meridional temperature gradient of the Tropical Atlantic can induce changes in precipitation. The response to vegetation cover is induced feedbacks between the ocean and the atmosphere (Braconnot, 2004). In a study focused on the Last Glacial Maximum (LGM) (21.000 ka BP), Kubatzki and Claussen (1998), concluded that the

climatic conditions of the SH are determined by ocean basins.

In this context, we selected four interglacials that have similarities in their orbital configurations, also because they portray the variability of climatic conditions and vegetation classification, namely, MH and the Marine Isotope Stages (MIS) 5e, 11c and 31 (1.072 million years ). Interglacials MH and MIS's 5e, 11c and 31 are marked by the strong meridional gradient of global SST anomalies that are induced by the seasonal distribution of insolation at the top of the atmosphere due to variation in long-term



orbital forcing (Berger, 1978, 2021; Smulsky, 2021). Therefore, for these four periods there is an analogous factor between them, which is the seasonal cycle of insolation being more intense in Northern Hemisphere (NH) and attenuated in SH due to the effect of the orbital parameter of precession (Berger, 1988).

MH a was a period with a weaker seasonal cycle in the SH in response to the variation of orbital parameters, the strong meridional gradient of the SST from the Tropical Atlantic, contributing to the displacement of the ITCZ to the North Atlantic (Shin et al., 2006), low $CO_2$ compared to the present day (Raynaud et al., 1993), in addition to having been important in the variation of the climate system (Berger, 1978), mainly in the modifications of the tropical vegetation of the SH (Braconnot et al., 2019).

MIS5e (127 kyr ago) also known as Eemian, experienced strong prolonged climate changes due to oceanic and atmospheric feedbacks in response to the seasonal insolation cycle (Yin and Berger, 2012; Berger and Loutre, 1991; Members, 2006; Siccha et al., 2015; Bard et al., 1990). Compared to global averages from SST paleoreconstructions, MIS5e was considered the warmest interglacial in the last 800 years (of PAGES, 2016). Although they have different nomenclatures, their definitions also differ. While MIS5e was based on marine oxygen isotopes, Eemian was derived from the change in global vegetation cover (Shackleton et al., 2003).

MIS11c (409 kyr ago) was also considered an interglacial with negative SST anomalies during the SH Austral Summer, but positive in the NH, from paleoreconstructions (Kandiano et al., 2017). Changes in equatorial cross meridional heat transport may have caused anomalous cooling in the South Atlantic during MIS11c (Riveiros et al., 2013). In a recent survey, Tzedakis et al. (2022) observed that the insolation forcing was weak when compared to other interglacials, however its prolonged duration was sufficient to reduce the Greenland and Antarctic sea ice sheets, causing sea level rise and smoothed release of $CO_2$.

MIS 31 was considered a super interglacial, being responsible for the melting of glaciers in Greenland when compared to the present day (Lisiecki and Raymo, 2005; Melles et al., 2012). Such changes in the climate system during MIS31 are a response to more extreme orbital combinations, that is, high values of precession and eccentricity were able to generate strong SST anomalies at high latitudes and reduction of sea ice during the boreal summer (Oliveira et al., 2017; Yin and Berger, 2012). MIS31 exhibits characteristics analogous to Eemian and MIS11 due to the reduction of the Antarctic ice sheet in response to the $\sim 5°C$ increase in SST in sectors of West Antarctica (Pollard and DeConto, 2009; Villa et al., 2012; Teitler et al., 2015). From simulations using a coupled model of intermediate complexity, an increase above $1°C$ was estimated during the northern summer (Justino et al., 2019).

In this paper, we investigate the dynamic vegetation response of tropical continental regions of the SH to SST variation of the different southern ocean basins, using two coupled CGCM models, the ICTP-CGCM (Speedy-Nemo) and the CCM3/IBIS. The study analyzes the interaction between atmosphere-continent-ocean, focusing on the parameters of precipitation, temperature and dynamic vegetation, generated from the simulations of the global dynamic vegetation model that were forced through the boundary conditions (SST) of individualized Speedy-Nemo basins, from the orbital forcing of interglacials MH e MIS's 5e, 11c e 31. To verify if the dynamic vegetation corresponds to the Mid-Holocene, we used the CCM3/IBIS simulation outputs to compare with the paleovegetation reconstructions of the BIOME6000 project (Prentice et al., 2000, 2011). The specific questions addressed here were the following: on the continental regions of the SH, specifically in monsoon areas, what is the





magnitude of the changes suffered by the dynamic vegetation? In addition to the orbital factor, could climate variation on a millennial scale related to OSH changes have impacted the vegetation cover of tropical wet and dry environments? How does the vegetation pattern respond to combinations of climate system changes and individual ocean basins?

## 2   Models and experiments

### 2.1   Models description

The present work applies the Global Coupled Atmosphere-Ocean Circulation Model from the International Center for Theoretical Physics (ICTP-CGCM), which consists of the global atmospheric climate model "SPEEDY" (versão 41), coupled to the Nucleus for European Modeling of the Ocean (NEMO) (Madec et al., 1998; Madec, 2008; Valcke, 2013). In computational terms, SPEEDY proofs to be effective in reproducing the main characteristics of the climate system of tropical and extratropical latitudes (Molteni, 2003; Kucharski et al., 2006; Justino et al., 2021). The atmospheric component runs at eight vertical levels in
T30 horizontal resolution (Kucharski et al., 2016) (See Table 1).

**Table 1.** Models description of this study

|  | ICTP-CGCM | NEMO | CCM3-IBIS |
|---|---|---|---|
| Caracters |  |  |  |
| Domain | Global | Regional / Global | Global |
| Spatial Resolution | 3,75° (lat) x 3,75° (lon) | 2° (lat) x 2° (lon) | 2,8° (lat) x 2,8° (lon) |
| Vertical Levels | 8 levels (925–30 hPa) | 31 levels (10–5000 m) | 18 levels (925–30 hPa) |
| Coordinates | Sigma | Hybrid-sigma Z | Hybrid-sigma |

To assess the impact of interglacial boundary conditions on vegetation changes, we employed the CCM3-IBIS coupled model. This model effectively captures the bidirectional interactions between land surface and atmosphere, playing a key role in characterizing vegetation distribution, ecosystem functionality, and climate-vegetation feedbacks under varying climatic and land-use scenarios (Foley et al., 2000).

The CCM3-IBIS is based on the atmospheric model CCM3 (Community Climate Model, version 3), developed at the National Center for Atmospheric Research (NCAR, USA). It has been integrated with 2.8° latitude and longitude (T42) resolution, and 18-level pressure-sigma hybrid coordinate system Kiehl et al. (1998). This model is coupled with a 2.6.4 version of the surface model known as the Integrated Biosphere Simulator (IBIS) Foley et al. (1996, 2000); Kucharik et al. (2000). IBIS is a global dynamic vegetation model that simulates changes in vegetation composition and structure in response to environmental
conditions Delire et al. (2002).

### 2.2   The experiments

The experiments consist of the control simulation (CTRL) conducted for the climate of the twentieth century (Figure 1), integrated 500 years after the creation of the ICTP-CGCM model, taking as the reference climate state. Four sensitivity



experiments have been performed to characterize intervals representing interglacial stages, MH, and MIS 5e, MIS 11c, and

MIS 31. For all experiments, the first 100 years of the simulation are not included in the current analysis of the results. These simulations apply boundary conditions characteristics of orbital changes and greenhouse gas concentration ($CO_2$, $CH_4$ and $N_2O$), as described in Coletti et al. (2015); Bereiter et al. (2015); Lüthi et al. (2008); Justino et al. (2017, 2019).

After achieving the quasi-equilibrium state for those individual runs, additional experiments have been carried out with CCM3/IBIS forced by the CTRL, MH, MIS5e, MIS11c, and MIS31 SST boundary conditions characteristics of these interglacial

eras. The glacials were conducted, but using individualized ocean basins, following the criteria used by Pereira et al. (2014), namely: the Tropical Indian Ocean (IND, 30°N-30°S and 40°E-110°W), Tropical Pacific (PAC, 30°N-30°S and 120°E-280°W) and Tropical Atlantic (ATL, 30°N-30°S and 300°E-360°W). It is worth mentioning that each experiment consists of one round of 50 years. However, the first 10 years are ignored and the last 40 years of each simulation are analyzed, in which the spin-up time of each experiment has been reached to obtain a more realistic representation of the climatology and vegetation. For more

detail, the characteristics of the rounds are summarized in Table 2, and the experimental design is summarized in the Fig. 1.





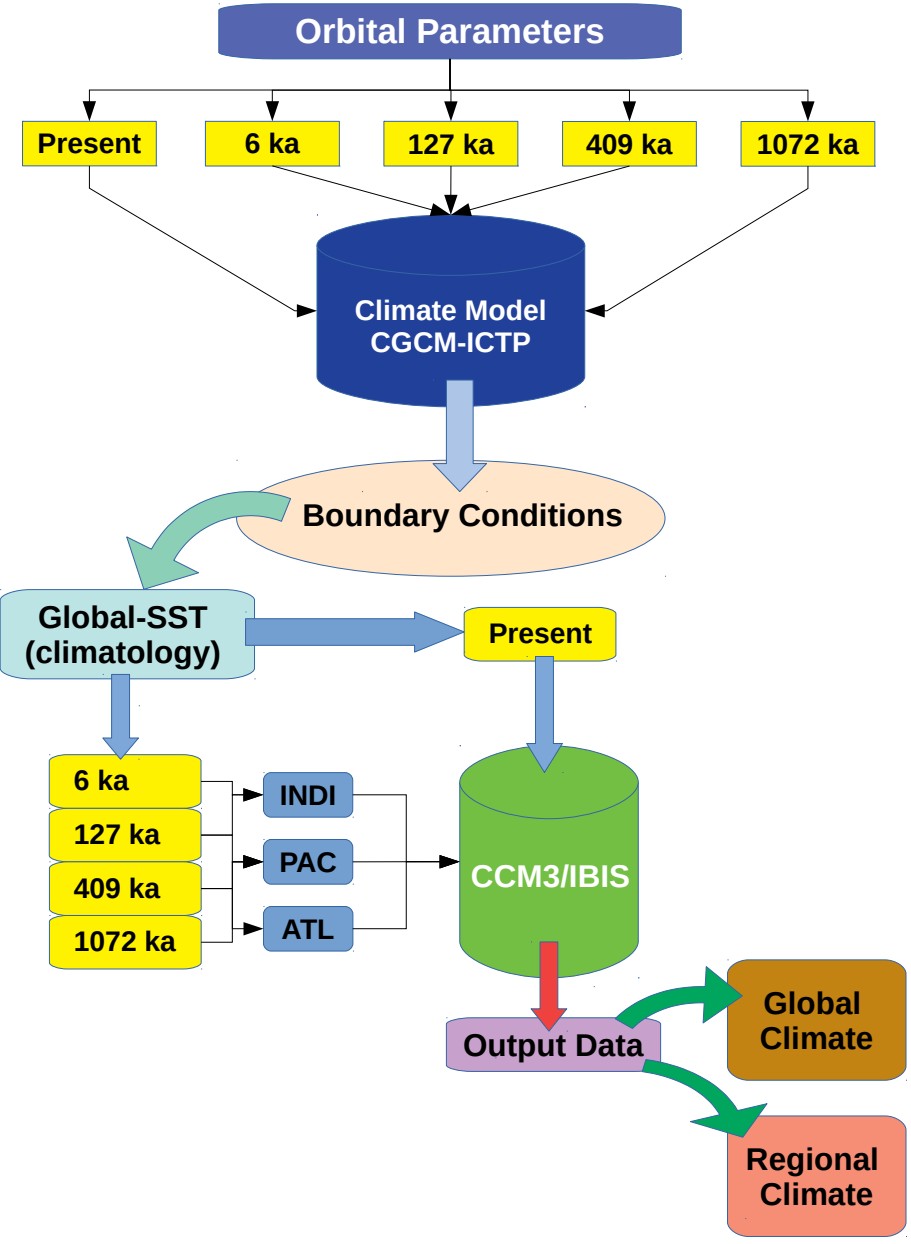

**Figure 1.** Schematic illustration of global model systems to study climate change past by CGCM-ICTP in the framework of the stages of the integration. The second model illustrates the stages from coupled model atmospheric-vegetation CCM3/IBIS. Orbital Parameters: Present; 6 ka = Mid-Holocene; 127 ka = MIS5e; 409 ka = MIS11c and 1072 ka = MIS31. INDI = Tropical Indian Ocean, 30°N-30°S and 40°E-110°W; PAC = Tropical Pacific, 30°N-30°S and 120°E-280°W and ATL = Tropical Atlantic, 30°N-30°S and 300°E-360°W. See text and tables for more detailed descriptions.



**Table 2.** Orbital configurations and greenhouse gases concentrations utilized in the CTRL, MH, MIS-5e, MIS-1c and MIS-31 experiments.

| Experiment | Date | $CO_2$(ppmv) | $CH_4$(ppbv) | $N_2O$(ppbv) | Ecc. | Obl. | Prec. |
|---|---|---|---|---|---|---|---|
| CTRL | Present day | 380 | 801 | 289 | 0.01671 | 23.438 | 101.37 |
| MH | 6 ka BP | 280 | 700 | 275 | 0.01868 | 24.105 | 0.87 |
| MIS5e | 127 ka BP | 287 | 724 | 262 | 0.03938 | 24.040 | 272.92 |
| MIS11c | 409 ka BP | 285 | 713 | 285 | 0.01932 | 23.781 | 265.34 |
| MIS31 | 1072 ka BP | 325 | 800 | 288 | 0.05597 | 23.898 | 289.79 |

Verification of the capacity of CCM3 / IBIS to reproduce the dominant land cover type under current conditions is based on comparison with the annual land cover map derived from the MODIS dataset (MOD12C1) for the year 2000 to 2020, with a resolution of 0.05 degrees (Hodges, 2002). This product contains the global vegetation classification scheme derived from the IGBP (International Geosphere Biosphere Programme).

# 3 Results and discussion

## 3.1 Surface temperature and precipitation differences

The surface temperature and precipitation differences averaged in time are shown in Figure 2. Temperature differences between the interglacial stages and CTRL demonstrate that negative anomalies are dominant features in the tropical ocean basins (Figure 2a-b-c), which are more pronounced in MIS 5e and 11c (Figure 2b-c) due to weak insolation over the SH in response to changes
in precession. It has to mentioned that across the Indian Ocean and in most of the MIS31 the SST are higher than in the CTRL experiment.

Turning to the pattern over land, positive temperature anomalies are shown over northern SAM, CAF and SAF, and northern Australia in MH, MIS5e and MIS11c. Despite the warmer ocean, MIS31 over the Amazon and Australia does not shown significant changes of temperatures with respect to the CTRL run (Figure 2d). However, the pattern over Africa is characterized
by warmer SH and colder NH, indicating the highly influential effect of precession.

Precipitation differences also shown in Figure 2 indicate an overall reduction in continental rainfall in the north-central SAM and south-central Africa, but with a slight increase over Australia, in the MH, MIS5e and MIS11c intervals (Figure 2e-f-g). Increased precipitation is also observed in Australia during the MIS31 (Figure 2h). The combination of climatic feedback such as the decrease in insolation and the reduction of the horizontal temperature gradient between the oceanic basins and continents
in the SH, lead to the weakening of the monsoons in Africa, Australia and SAM, hampering the transport of water vapor from the ocean to the mainland during the rainy season. These changes in precipitation associated with the decrease, in turn, tend to reduce evapotranspiration, and consequently favor a positive feedback. The annual reduction in precipitation over the tropical region of the SAM may be associated with the weakening of evapotranspiration which in turn leads to the desintensification of deep convection in the Amazon (Wright et al., 2017). In summary, the effects of the rainfall pattern on Amazonia during
interglacials depend on the magnitude of the differential warming between continent and ocean, once this warming becomes



**Figure 2.** Spatial distribution of annual mean surface temperature difference and annual mean precipitation difference between interglacials experiments compared to SST and precipitation from CTRL; MH minus CTRL (a,e), MIS5e minus CTRL (b,f), MIS11c minus CTRL (c,g) and MIS31 minus CTRL (d,h). Black dots represent correspond to statistically significant anomalies at the 95% confidence interval.

attenuated, the ascending branch tends to be weakened, carrying less water vapor to the high levels of the troposphere and reducing the release of latent heat through condensation.

## 3.2 Changes in cover land

The climate response to SST changes revealed that precipitation plays a key role in land cover distribution. Figure 3 shows
the spatial distribution of the aridity index, which may be treated as an indication of dry climates over the Africa, SAM and the





Australia sectors. The aridity index ($I_{UNEP}$) is evaluated in this study to identify regions that are affected by water stress, as provided by the relationship between precipitation and evapotranspiration. The $I_{UNEP}$ is defined by the equation

$$I_{UNEP} = \frac{P}{ET_0} \tag{1}$$

where, $P$ is the average annual rainfall provided by CCM3/IBIS, and $ET_0$ is the annual potential evapotranspiration, calculated
using the Thornthwaite (1948) and adjusted by Penman (Penman, 1948). The spatial distribution adopted in this study is based on the classification of the aridity index similar to the current climate (Feng and Fu, 2013). Dry and wet climates are defined by $I_{UNEP}$ lower than 0.65, hyperarid ($I_{UNEP} < 0.05$), arid ($0.05 \leq I_{UNEP} < 0.2$), semi-arid ($0.2 \leq I_{UNEP} < 0.5$), dry-subhumid ($0.5 \leq I_{UNEP} < 0.65$) and humid ($I_{UNEP} > 0.65$).

In the CTR run, the CCM3/IBIS model is able to reproduce arid and semi-arid areas located in tropical and subtropical regions,
such as Southern Africa, Australia and Northeast Brazil (NEB) and Southern SAM (Figure 3). The most arid lands are located in the Sahara/Sahel and central Australia, as expected. By analyzing the $I_{UNEP}$ delivered by past climate through differences in the spatial distribution between the experiments and CTRL run, it is noticed that positive temperature anomalies ranging from 1-3°C over the continents (Figure 2), favoring the expansion of dryland over the Congo region, Southern Africa, Indonesia and much of the Amazon during MH, MIS5e and MIS11c (Figure 3b-c-d). The prolonged drought over the tropical region
can cause changes in large-scale meteorological systems such as the ITCZ. From supplementary materials, Nilsson-Kerr et al. (2021) disponibility a summary of the available global compiled hydroclimate proxy data for MIS5e. A comparison between CCM3/IBIS experiments and hydroclimate proxy indicates a dry continent over the Amazon region in North Brazil. Therefore, the main spatial patterns of aridity are well established in response to decreased insolation in southern hemisphere summer during MIS5e.



**Figure 3.** Aridiy index to CTRL (a), MH minus CTRL (b), MIS5e minus CTRL (c), MIS11c minus CTRL (d) and MIS31 minus CTRL (e).

In addition to the macroscale effect, there is also a decrease in local convection due to subsidence, which may cause suppression of precipitation not only locally, but also in remote regions. Despite these changes being on past distant intervals and representing the natural variability of the climate system, studies aimed at understanding deforestation through coupled modeling Nobre et al. (2009), suggest that the reduction of rainfall over the Amazon depends on a set of complex factors, related



to the weakening of the Equatorial Pacific temperature gradient, reduction of deep convection in the Amazon and changes in land
use, which favors the weakening of the ascending branch of the Hadley Cell in the latitudes between 10°S-0 (Supplementary
Figure S2). MIS31 does not show substantial changes for the aridity index (Fig. 3e). However, dryer condition may be found
over South America and subtropical Africa.

It might be expected that these changes in the hydrological cycle as reproduced by the $I_{UNEP}$, lead to changes in vegetation
patterns during the past interglacial eras. Higher continental temperatures increases the potential evapotranspiration that can
reduce soil moisture, if a lack of precipitation takes place. Theoretically, dry soil conditions will result in the presence of much
sparse vegetation affected by enhanced seasonal changes.

### 3.3 Global patterns of vegetation

Figure 4 shows the distribution of observed vegetation from the MODIS data and the dynamic vegetation simulated by the
CCM3-IBIS model based on the CTRL simulation. It has to be mentioned that MODIS vegetation applies a mathematical
algorithm to convert radiances in physical vegetation properties, and delivers mean conditions to the year 2010. The CCM3-IBIS
model, on the other hand, simulates the potential vegetation related to mean climate conditions, and does not include the
anthropogenic influence associated to deforestation. Thus, regions with evergreen forests in South America and Africa according
to MODIS cover small areas with respect to CCM3-IBIS.

Overall, CCM3-IBIS is able to reproduce the main features of the African, Australian and South American (SAM) biomes
when compared to the MODIS dataset. At the regional scale differences are noted. The simulated vegetation over Central Africa
shows grassland, evergreen tropical and deciduous tropical forests, while the observed vegetation map shows savanna and mixed
forest in parts of the Congo region. The model simulated tropical evergreen forest across the Indonesia region, corroborating
the satellite-based vegetation. The land cover over Australia slightly differs from observed vegetation; MODIS shows most
mixed and deciduous forests and savannas, CCM3-IBIS simulates boreal evergreen forest, mixed forest and savannas; over the
northern portion of Australia the simulated landcover is dense shrubland, but the observed vegetation map shows savanna and
grassland. In South America, the dominant region of the tropical evergreen forest of the Amazon is simulated, in some areas
over Venezuela and Colombia, the MODIS dataset shows savanna, but the CCM-IBIS reproduced tropical deciduous forest.

Turning to interglacials experiments, the South America vegetation experiences significant changes during MH, MIS5e and
MIS11c (Fig.4 c-d-e), as a consequence of the drop in temperatures and air humidity over the southern tropical region, with
exception of MIS31 (Fig.4 f). During the MH, MIS's 5e and 11c dominant biomes such as evergreen and deciduous forest over
the Amazon have been replaced by savannas and grasslands, mainly in the west/south of Amazon, which are sectors affected by
the South American monsoon, as well as reduced atmospheric $CO_2$ levels during those past intervals weakens the physiological
effect related to fertilization. Based on simulations to reconstruct the global carbon cycle, Schurgers et al. (2006) demonstrated
that large dominance of areas covered by savanna over Amazon in response to a dry climate during the Holocene and Eemian.



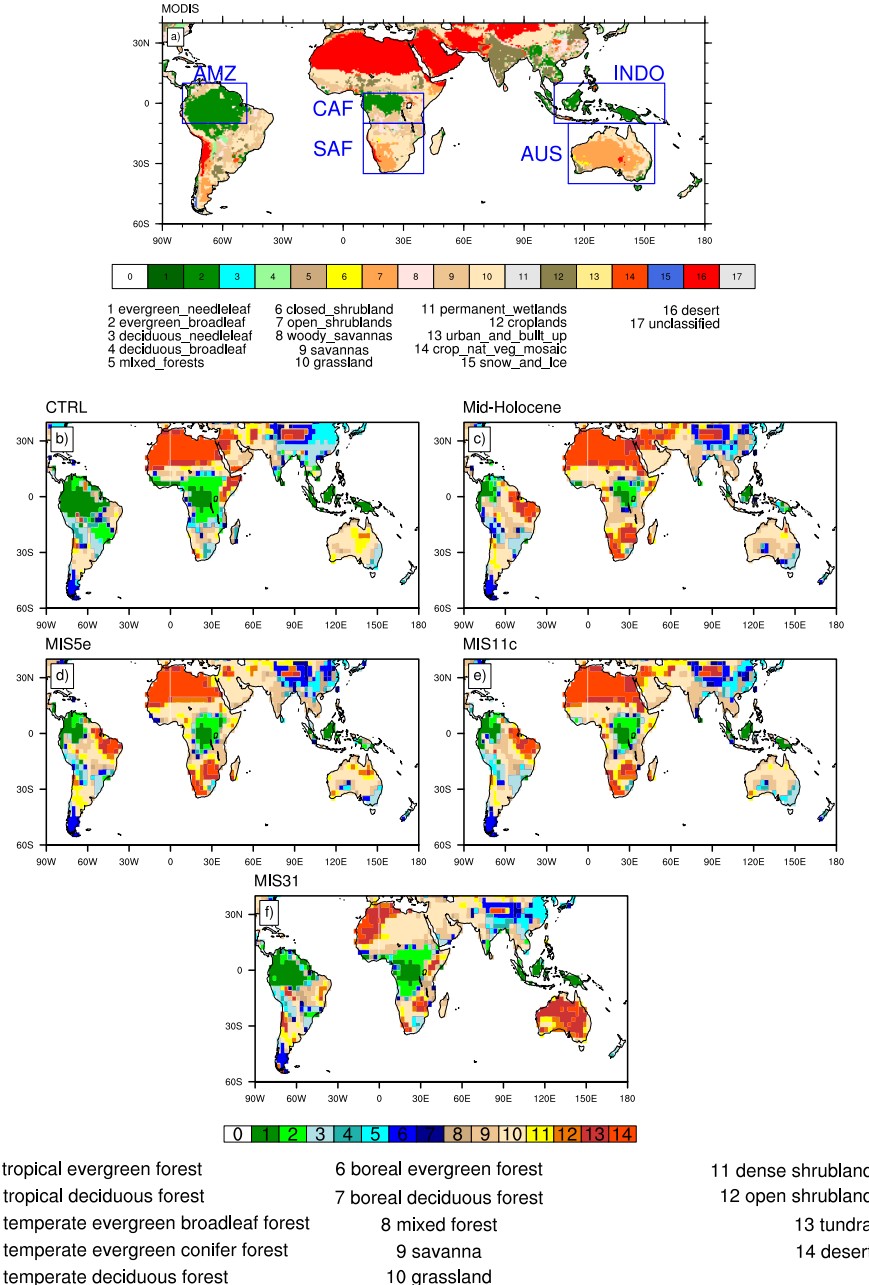

**Figure 4.** Land cover map derived from MODIS data (MOD12C1) (a), and defined in this study at T42 spatial resolution from CTRL by CCM3/IBIS (b), MH (c), MIS5e (d), MIS11c (e) and MIS31 (f), experiments. Regions delimited in blue boxes refer to domains in which percentage classification vegetations, precipitation e temperature are averaged, as analyzed in this paper are shown; AMZ: Amazonia (10°N-10°S and 50°W-80°W); CAF: Central Africa (5°N-10°S and 10°E-40°E); SAF: South Africa (10°S-35°S and 10°E-40°E); INDO: Indonesia (10°N-10°S and 105°E-160°E); AUS: Australia (10°S-40°S and 112°E-155°E).



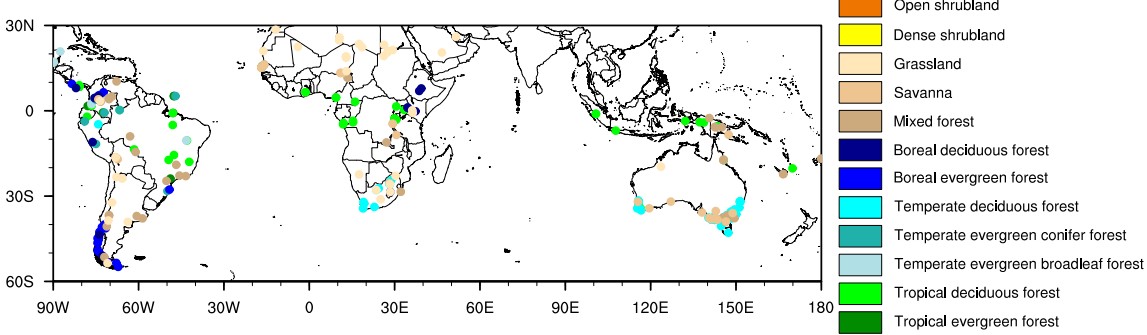

**Figure 5.** Biome mid-holocene palaeovegetation data from BIOME 6000 (Prentice et al., 2000). Coloured circles represent vegetation classification each, respectively. See Table S2 in Supplement.

Ledru et al. (2001); Ledru (2002); Prado et al. (2013) based on pollen inferences found an expansion of savanna and pasture over eastern Amazon and central Brazil between the Holocene and Middle Holocene periods.

Other extreme scenario is also shown in the NEB, where the caatinga biome is replaced by deserts in the MIS5e, MH and MIS11c interglacials (Fig.4 c-d-e). This suggests that the shift from the caatinga biome to the desert may be associated with a dry climate and reduced atmospheric $CO_2$ concentration. These changes can be attributed to the intensification of feedback mechanisms through the anomalies that generate diabatic cooling, as discussed in past studies (Zebiak, 1986). For southern Brazil, the model was able to simulate grass vegetation that may be related to the pampa, as well as a much smaller arboreal fraction over the Southeast when compared to CTRL. Overall, our results corroborate the results of previous studies such as Behling et al. (2005), where it was found that pollens from São Francisco de Assis, Rio Grande do Sul, indicate grasses with forest refugees, associated with hot and dry climates during the Holocene.

Changes in vegetation dynamics during MIS31 are also remarkable. For example, Australia was totally desertified, part of the Sahara changed from desert to savannah, and tropical forest increase in Tropical Africa. The Amazon perennial forest remains dominant as simulated by the CTR run, however, savanna begins to expand between southern Amazonia and northern central South America (Fig.4f).

Figure 5 shows vegetation reconstruction from the BIOME6000 project for MH. As verified and discussed in Figure 4 c, our simulation matches well with the grassland domain and the tropical deciduous forest between Central and Southern Africa. In addition, good correspondence is also found with savanna expansion across the Congo region and Australia. The inferred pollen data do not show the retreat of the tropical evergreen forest, but the BIOME6000 dataset covers a small region. Therefore it is not feasible to draw considerations for the entire Amazon domain.



## 3.4 Change in vegetation areal coverage

Analyses for changes in vegetation types during the interglacials demonstrated that parts of the CAF region under CTRL conditions are covered by approximately 35% evergreen tropical forests, 34% deciduous forests, mixed forests 3% and 9% grasslands (Table 3, Fig. 6 a). During MH, MIS5e, and MIS11c, there is a systematic reduction of forests by up to 60%, 50%, while there is an increase of grassland in particular during MH. Savannas that is almost absent in the CTRL climate over the CAF region, experience increases for all interglacial stages, due to enhanced climate seasonality with well defined months

governed by low precipitation. Paleohydrological analyzes indicate that the Sahara/Sahel regions during the MH large extent of grass and savannah vegetation (Lézine et al., 2011) were present in agreement with the simulated results shown in Table 3.

Over the SAF domain, the CTRL delivers 11% of perennial forest and 14% of the Atlantic forest (Fig. 6 b). There were no discrepant changes among the interglacials, where the dominant savanna and grassland vegetation remained conserved with small variations. It should be stressed that climate conditions during the MH, MIS5e, MIS11c does not support evergreen forests

(Fig. 2 and Table 3). The MIS31 differs from other interglacial and shows very similar features as delivered by the CTRL. It is noticed that precipitation changes are small which are accompanied by lower temperatures.

In our CTRL simulation Indonesia (INDO) was dominated by tropical evergreen forest (Fig. 6 c), thus it reproduced the observed vegetation very well, although there was a reduction in the MH interglacials (53%), MIS5e (50%), MIS11c (34%) and MIS31 (10%), being replaced by Mixed forest during MH (21%), MIS5e (14%) and MIS11c (21%), except for MIS31.

The reproduction of the dominance of the savanna and grasslands in Australia (Fig.6 d), in the CTRL experiment, was similar to the vegetation observed. Despite having successfully predicted dominant biomes, CCM3 / IBIS was unable to reproduce the tropical evergreen forest cover over southeastern Australia, but was replaced by 9% temperate evergreen broadleaf forest. Even so, the differences are few between the biomes of the paleoclimatic periods and the CTRL simulation, conserving the Temperate evergreen broadleaf forest over the same region of Australia in the MH, MIS5e and MIS11c, except in the MIS31, where there

was a reduction of 82% of the savanna and 78% of pasture, while it showed coverage of 16% of tundra and 48% of desert.

CCM3 / IBIS overestimated the biome of Tropical Evergreen Forest (Amazon rain forest) by 12% in northern SAM compared to observed vegetation, but warm paleoclimatic periods showed a reduction of 76% (MH), 58% ( MIS5e), 64% (MIS11c) and 9% (MIS31), compared to CTRL experiment. However, there was an increase in savanna and pastures, mainly in the periods MH (81% and 95%), MIS5e (80% and 93%) and MIS11c (75% and 94%).





**Figure 6.** Bar chart of percent (%) change from biome type of the dominant classifications over each study region. (a) CAF, (b) SAF, (c) INDO, (d) AUS and (e) AMZ.





**Table 3.** Percent change in the number of pixels from biome type

| Biome type | CTRL | MH | MIS5e | MIS11c | MIS31 |
|---|---|---|---|---|---|
| | | | CAF | | |
| Tropical evergreen forest | 35% | 14% | 24% | 20% | 38% |
| Tropical deciduous forest | 34% | 17% | 22% | 19% | 24% |
| Mixed Forest | 3% | 3% | 5% | 8% | 8% |
| Savanna | 0.1% | 30% | 17% | 13% | 6% |
| Grassland | 9% | 15% | 12% | 13% | 3% |
| | | | SAF | | |
| Tropical deciduous forest | 11% | 0.1% | 0.4% | 0.1% | 7% |
| Temperate evergreen broadleaf forest | 14% | 1% | 1% | 1% | 9% |
| Temperate evergreen conifer forest | 9% | 1% | 2% | 3% | 4% |
| Savanna | 17% | 19% | 15% | 16% | 13% |
| Grassland | 18% | 17% | 12% | 18% | 22% |
| | | | AUS | | |
| Tropical evergreen forest | 0 | 0 | 0 | 0 | 0 |
| Savanna | 11% | 29% | 18% | 21% | 2% |
| Grassland | 50% | 48% | 47% | 50% | 11% |
| Tundra | 3% | 0% | 7% | 1% | 16% |
| Desert | 0% | 1% | 1% | 0% | 48% |
| | | | IDA | | |
| Tropical evergreen forest | 100% | 47% | 50% | 66% | 90% |
| Tropical deciduous forest | 0 | 4% | 12% | 0 | 0 |
| Temperate evergreen conifer forest | 0 | 12% | 0% | 0 | 6% |
| Mixed forest | 0 | 21% | 14% | 21% | 0 |
| | | | AMZ | | |
| Tropical evergreen forest | 75% | 18% | 31% | 27% | 68% |
| Tropical deciduous forest | 8% | 7% | 8% | 5% | 10% |
| Mixed forest | 2% | 0.1% | 0.1% | 3% | 2% |
| Savanna | 3% | 16% | 15% | 12% | 4% |
| Grassland | 0.1% | 24% | 15% | 18% | 1% |

## 3.5   Climate response under individualized ocean basin changes

Precipitation responses over continental areas of Africa, Australia and South America, caused by changes in individual ocean basins such as PAC, ATL and INDI, during interglacials are shown in Figure 7. For the MH, the results indicate that the PAC has no association with the precipitation of the AFR, AUS and SAM sectors (Fig. 7a). However, the ATL was the basin that most influenced the significant reduction in SAM and AFR precipitation (Fig. 7b), although the INDI caused a reduction in southern Africa (Fig. 7 c). As shown in Fig. 7d-f, g-i e j-l, the PAC induces a reduction in precipitation only in South Africa, except for MIS31, where southeastern SAM and Australia also had similar responses. For the three MIS interglacials 5e, 11c and 31, the SAM and AFR are totally dependent on the ATL. This suggests that much of the cooling of the equatorial Atlantic causes a





reduction in precipitation, as the ITCZ migrates to the Boreal Hemisphere in response to the variation of this ocean basin and acting on the photosynthetic processes of the Tropical Forest. Under climatological conditions, for all interglacials, INDI has

simulated significant rainfall reduction in parts of southern Africa and Australia.

When calculating annual averages of precipitation over the areas of CAF, SAF, AUS, IND and AMZ, strong connections were found with variations in SST (see precipitation values for climatological conditions on Figure S3 in Supplement). In the CAF region, during the MH, MIS5e, and MIS11c periods, average annual precipitation decreased by approximately 18%, 15%, and 15.5% (-0.71, -0.57, and -0.59 mm/day), respectively. (Fig. 8a-f-k). Although the Congo region is strongly influenced by

the ITCZ during austral summer and fall, this negative variation in precipitation from these three interglacials is a response to the anomalous cooling of the ATL SST, as cold water reduces convection and increases stratus cloud formation, causing the ITCZ to migrate to the Boreal Atlantic Ocean. Despite changes in precipitation during the closest interglacials, precipitation in MIS31 turns out to be very close to climatology (CTRL) when the model is forced by the ATL and PAC experiments (Fig.8j). The reduction of annual precipitation was also shown on the SAF with 30%, 29%, 26% and 11% (-0.65, -0.63, -0.56 and -0.25

mm/day) in MH, MIS5e, MIS- 11c and MIS31 (Fig. 8a-d-g-j). This water deficit is associated with changes in IND SST patterns, since the differences between $IND_{MH}$ minus CTRL simulations show reductions of 27% as well as 24% for $IND_{31}$ minus CTRL; MIS's 5e and 11c, showed suave rainfall when the model was forced with the CAP's SST anomalies.



**Figure 7.** Continental distribution of precipitation (mm/day) anomalies for respective rows and columns representing the oceanic basins regions forced in MH (a-b-c), MIS5e (d-e-f), MIS11c (g-h-i) and MIS31 (j-k-l) experiments. Black dots represent areas where the results are statistically significant (at a 95% confidence level).





**Figure 8.** Areal averaged precipitation (mm/day) CAF, SAF, AUS, IND and AMZ, region for the SST anomaly sensitivity experiments.





The increase of 38%, 27% and 32% (+0.81, +0.58 and +0.69 mm/day) is shown over AUS, in MH, MIS5e and MIS11c, in relation to climatology (Fig. 8c-h-m), although MIS31 has a reduced average annual rainfall of 32% (-0.68 mm/day) (Fig.

8r), these changes occurred in all interglacials are associated with the SST anomalies that were forced by the PAC basin. The Indonesia region (INDO) showed a weak increase when compared to the climatology of annual precipitation, around 1%, 2% and 2.5%(+0.08, 0.13 and 0.17 mm/day) in MH, MIS5e and MIS11c (Fig. 8d-i-n), except for MIS31, where there was a 5% reduction (-0.39 mm/day) (Fig. 8s).

The average annual precipitation is lower over the AMZ region when we observe the periods MH, MIS 5e e 11c (Fig.8e-j-o),

where there was a reduction of 30%, 20% and 25% (-1.7, -1.16 and -1.4 mm/day). However, a weak increase of almost 2% (+0.08 mm/day) is observed in MIS31 (Fig.8t). These changes in the hydrological cycle over the AMZ are due to variations in SST in the ATL basin, because when quantifying the differences between the $ATL_{MH}$, $ATL_{5e}$ and $ATL_{11c}$ experiments with CTRL, there were decreases of 32%, 27% and 35%. In the analysis of the regional mean temperature of the AMZ, the experiments MIS-5e, MH and MIS-11c exhibit heating of almost 3°C, which is also attributed to the ATL forcing (see Figure

S4 in the Supplement). This suggests that oceanic energy transport leads to an interhemispheric shift of ITCZ (Marshall et al., 2014), mainly to the hemisphere, providing greater exchange of energy between the ocean and the atmosphere (Chiang and Friedman, 2012). However, MIS31 shows similarity to current temperatures (Fig.8h), in addition to the contribution of the SST forcing through the PAC basin.

## 4 Conclusions

Although the orbital parameter has a strong influence on the natural variability of the planet, especially on the precipitation variable, this factor is not the only factor that explains the sensitivity of the CCM3/IBIS model.

In fact, changes in the SST of the ocean basins were affected by radiative forcing due to the precessional factor during interglacials, resulting in the displacement of the ITCZ to the NH, in addition to feedbacks from the climate system (Chiang and Bitz, 2005). During the austral summer, the annual precipitation rate is reduced over South America, southern Africa, and

northern Australia in response to cold SST anomalies and high precession. Although our results do not provide an in-depth analysis of the dynamic mechanisms associated with ITCZ, our interpretation is that the reduction in the flow of moisture from the Atlantic to the Amazon caused an imbalance in the annual precipitation rate of the AMZ, and consequently made it a climate dry and prolonged, leading to the reduction of the tropical evergreen forest biome to the expansion of savannas and grasslands.

The high precession and obliquity values during summer and winter were not enough to strengthen the South American

monsoon (Supplementary Figure S1 a-d), although there was an induction of an anomalous low locally over Amazonia during the MH and MIS 5e and 11c, the moisture transport inferred by the North Atlantic was not sufficient to maintain deep convection. The precipitation decrease coincides with the anomalous high pressure center between the South Atlantic basin and the South American continent induced intense cooling between the Equatorial and Southern Atlantic. Under all these conditions mentioned, even though the studied periods are of remote time scale, that is, millennium, the natural variability without anthropogenic

influence and stabilized $CO_2$ levels, which leads us to suggest that the model CCM3/IBIS was able to produce a "dieback" of the





Amazon forest (Parry et al., 2022; Malhi et al., 2009), that is, the vast region inhabited by vast perennial vegetation tends to extinction due to drastic reductions in precipitation and long periods of drought. The opposite effect can be observed between the Sahara/Sahel and Congo regions during boreal summer (Supplementary Figure S1), where a wide anomalous low is induced during all interglacials, leading to increased continental humidity due to humid winds. from the North Atlantic.

Although CCM3/IBIS performs well in reproducing the current climate, the lack of pollen paleo-reconstruction data makes it difficult to analyze interglacial periods before MIS5e. What we can conclude is that the dynamic vegetation distributions that we analyzed are well related to SST changes. However, we cannot explain that the continental tropical rains that occur over the SH monsoon sectors are totally associated with SST changes. Roberts and Hopcroft (2020) suggest that changes in heat transport that occur in the ocean are more associated with climate balance, although there is a link between the location of the ITCZ and

atmospheric feedback. However, in a recent study, Venancio et al. (2022) through paleoecoconstructions suggest that the main causes of ITCZ migration during abrupt MIS5 events were modulated by oceanic heat transport and variations in NH ice cover.

     The impact of extreme aridity was sufficient to cause extreme drought in the tropical regions of Congo and Amazonia during the MH, MIS5e, and MIS11c interglacials. The analysis presented here suggests that the equatorial convection of these sectors depends on the sources of moisture from the evapotranspiration and advection processes of humid air from warm SSTs. However,

the effects of the precipitation deficit on the continent causes a long-term retraction of the perennial vegetation, since the vertical water vapor transport became attenuated due to the decrease in local evapotranspiration, and therefore, these changes made the driest and most extreme conditions. Drier conditions for a tropical forest can result in the adverse effects of evapotranspiration that are critical to the distribution of convective activity. Although our simulations show an increase in temperature in regions over the Amazon rainforest, it was not possible to intensify the convective cell of precipitation.

The relationship between $NIÑO_{34}$ and precipitation over the study regions was relatively weak and not significant across all interglacials (see Table S1 in the Supplement). Although there is no influence on the interannual variations of the precipitation pattern, it is worth mentioning that the CCM3/IBIS model was forced from the climatological boundary conditions, and therefore, this may not accurately exhibit the significant relationship, which suggests that oceanic forcing tropical Pacific in ENSO years is not entirely clear when it is correlated with SH continental precipitation. On the other hand, as shown in this study, the SST of

oceanic basins play an important role in determining the continental climatological pattern, as well as in the internal mechanisms of precipitation and in the pattern of vegetation. Although tropical regions are influenced both by climate and vegetation in response to Pacific and Atlantic SSTs, Nobre et al. (2009) combined simulations for scenarios with changing vegetation cover from a coupled model to investigate whether vegetation can play a role in modulating the ocean. The collaborators verified that in the scenario of forest removal there was a weakening of the deep convection of the Amazon, following a series of events,

such as a reduction in the trade winds and meridional circulation. In a recent study focused on the current period, Towner et al. (2021) showed that the response of the hydrological cycle over Amazonia depends on the ENSO phase, but overall it can play an important role in vegetation, i.e. the probability of savanna occurrence and tropical deciduous rainforest are less related to Equatorial Pacific anomalies.

     Our findings offer valuable insights into how SST variations in distinct ocean basins influence tropical precipitation patterns

and dynamic vegetation responses during past interglacial periods. What we want to express is that without forcing these ocean



basins it would not be possible to highlight the sensitivity of the CCM3/IBIS model to the response of the vegetation pattern. Despite the absence of these records, the classification of the dynamic vegetation of MH, MIS5e and MIS11c were very similar.

*Code availability.* All data are available upon request to the Corresponding Author.

*Data availability.* All data are available upon request to the Corresponding Author.

*Code and data availability.* All data are available upon request to the Corresponding Author.

*Author contributions.* Dr. Carlos Gurjão and Dr. Flávio Justino designed the study, performed data processing, and plotting, and wrote large portions of the manuscript. The authors would like to thank Fabrcio Murta for input on the CCM3 / IBIS model installation, and also thank Dr. Mônica Senna and Dr. Marcos Pereira for contributing with the simulations and support. The all simulations were performed at the Federal University Viçosa.

*Competing interests.* The Authors declare that there is no conflict of interest.

*Acknowledgements.* The authors want to thank the funding support of CAPES to project Research Group on Climate Interaction (InteraC), and funding was provided by CNPq funding 303882/2020. The Simons Foundation has provided funding to support the visits of Flávio Justino and Carlos Gurjao to The Abdus Salam International Center for Theoretical Physics. Fred Kucharski contributed to the interpretation of the model and the overall results. The authors thank Dr. Gabrielle Pires for your helpful remarks. This publication is part of the Research Group
on Climate Interaction (InteraC).



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
