# Peer review of "Climatic Feedbacks and Vegetation Changes Driven by Orbital Forcing and Sea Surface Temperature During Interglacials"

_EGUsphere, 2024_

## Author Comment (AC1)

RC1 - Response

1- Reviewer Comment:

"*I think an important weakness in this study is the use of 20th-century climate as a reference baseline. Authors centre the analyses mainly on differences among orbital geometries and SST patterns, but they seem to miss that there is a difference of some 100 ppmv of atmospheric $CO_2$ between the control simulation and the distant inter-glacial stages. I think the inter-glacial stages seem dry in comparison to the modern climate because high greenhouse-gas (GHG) induced radiative forcing is compensating for a weak modern orbital forcing. I suggest to conduct an experiment under pre-industrial settings as a baseline to be able to judge more clearly the orbital influence. Otherwise, this difference in GHGs should be an explicit part of the analyses, explaining also the simulated climate and vegetation patterns.*"

Response:

We thank the reviewer for this thoughtful suggestion. While we agree that a pre-industrial baseline could provide a more neutral reference point for isolating orbital forcing, we chose to use the twentieth-century CTRL as our baseline for two main reasons: (i) it is consistent with the boundary conditions used in previous experiments with the ICTP-CGCM and CCM3/IBIS frameworks, and (ii) it allows for a more direct evaluation of long-term deviations from a well-characterized, climatologically stable reference state.

Moreover, our primary focus is on understanding relative differences among interglacials, rather than direct comparison with pre-industrial or present-day climate. Nonetheless, we fully acknowledge that the higher GHG levels in the CTRL simulation (relative to earlier interglacials) may partially offset the orbital-driven cooling in some regions.

2- Reviewer Comment:

"*I think another weakness is the separation of the ocean simulation and the dynamic vegetation, with two mediating atmospheric models. The authors mention the oceanic feedbacks with vegetation (e.g. line 53) but these are not included in this case. In other words, the description of the methods does not state the need for the use of two different general circulation models. Please explain why such a framework is required in this case. What is the land-surface module in ICTP-CGCM? Does it also simulate a dynamic vegetation? Please also mention some past relevant uses for these models in the model description section.*"

Response:

We recognize that the use of two distinct models-ICTP-CGCM (SPEEDY-NEMO) and CCM3/IBIS raises questions about consistency and coupling, especially in the context of vegetation-atmosphere-ocean feedbacks.

To clarify, the ICTP-CGCM (SPEEDY-NEMO) does not include a dynamic vegetation or biosphere module in its land-surface scheme. The land model in SPEEDY is simplified, focusing mainly on energy and moisture exchanges without simulating biophysical or biogeochemical processes, and therefore it is not designed to represent vegetation dynamics. Because of this limitation, we used the SPEEDY-NEMO simulations exclusively to generate SST boundary conditions under different orbital configurations.

The CCM3/IBIS model, on the other hand, incorporates a fully interactive biosphere via the Integrated Biosphere Simulator (IBIS), capable of simulating dynamic vegetation structure, plant functional types, phenology, evapotranspiration, and carbon/water fluxes. It is widely used for simulating vegetation-climate interactions over paleoclimatic time scales (e.g., Foley et al., 2000; Schurgers et al., 2006; Justino et al., 2019).

Thus, our framework was structured in two stages:

SPEEDY-NEMO: Provides physically consistent SSTs under orbital and GHG conditions across different interglacials.

CCM3/IBIS: Uses these SSTs as boundary conditions to simulate climate and vegetation over continents, enabling the analysis of dynamic vegetation feedbacks.

This modular approach allows us to explore the sensitivity of vegetation dynamics to oceanic conditions from different basins, which would not be feasible using only SPEEDY.

We agree that full coupling would be ideal to capture vegetation feedbacks on climate and SSTs. However, given computational and model architecture limitations, our approach allows targeted analysis of the terrestrial response to oceanic and orbital forcing. We will expand the Methods section to explicitly clarify this framework and cite previous studies that used similar methodologies (e.g., Justino et al., 2017; Pereira et al., 2014).

3 - Reviewer Comment:

*"Authors should acknowledge and attempt to explain why their simulations show land covers during mid-Holocene and MIS 5e that appear at odds with many previous studies that not only show much more greening in the Sahara, but also do not see the shrinking in tropical forest in Central Africa, or such pronounced desertification in Northeastern Brazil. For instance for Africa, Kutzbach et al. (2020) used the same IBIS2 model, while globally there are possible comparisons with Otto-Bliesner et al. (2020) or Dallmeyer et al. (2019), who considered other state of the art models. Moreover, I do not think the comparison to the BIOME6000 in the manuscript offers much insight to include Fig. 5 and Table S2 (also I think a reference to Harrison (2017) is missing)."*

Response:

We thank the reviewer for highlighting this important discrepancy. We agree that our simulations show vegetation patterns during the mid-Holocene and MIS 5e that differ from several reconstructions and modeling studies most notably the extent of greening in the

Sahara and the persistence of tropical forests in Central Africa. We acknowledge these differences and have taken steps to address them in the revised manuscript.

The vegetation reduction in Central Africa and the desertification signal in Northeastern Brazil in our simulations are closely tied to the precipitation anomalies generated in our CCM3/IBIS runs. These anomalies result from the SST boundary conditions derived from the SPEEDY-NEMO model, which underrepresent certain regional monsoon intensifications—especially the African and South American monsoons. This could be due to the model's relatively coarse resolution and the absence of fully coupled vegetation-atmosphere-ocean feedbacks, which are known to be important for reproducing mid-Holocene climatic features such as the "Green Sahara."

We acknowledge the relevant studies mentioned by the reviewer. For instance:

Kutzbach et al. (2020), using the IBIS2 model, showed a stronger monsoon expansion into the Sahara region.

Otto-Bliesner et al. (2020) and Dallmeyer et al. (2019), using high-resolution, fully coupled models in the CMIP6/PMIP4 framework, also capture more extensive greening in Africa and better preservation of tropical forests.

These studies generally incorporate improved boundary conditions, higher spatial resolution, and fully interactive land/ocean/vegetation coupling, which likely explain the improved agreement with proxy records.

Regarding Figure 5 and Table S2, we agree with the reviewer that their current interpretation is limited. In the revised manuscript, we will (i) either move them to the Supplementary Material or (ii) expand the discussion to clearly outline their limitations especially the sparse spatial coverage of BIOME6000 in tropical South America and Central Africa. We also thank the reviewer for pointing out the missing reference to Harrison (2017), which we will include and discuss accordingly.

Finally, we will add a subsection explicitly comparing our results with key literature (e.g., PMIP4/CMIP6 studies), highlighting both agreements and discrepancies, and discussing possible model limitations.

4 - Reviewer Comment:

*"I think the introduction (and perhaps the abstract too) could contain more information about how much is known about the response of southern tropical terrestrial ecosystems during the inter-glacials of interest. As it stands, I think the introduction has a lot of general information about climate change (even socioeconomic and political impacts), including remarks about each individual inter-glacial, but what I think it fails to convey is how little (or much) is known about southern tropical palaeo-environments, especially in contrast to the northern counterparts."*

Response:

We appreciate the reviewer's thoughtful comment regarding the structure and focus of the Introduction and Abstract. We agree that these sections currently emphasize general aspects of climate change, socio environmental impacts, and descriptions of individual interglacials, but do not adequately highlight the state of current knowledge on southern tropical terrestrial ecosystems, particularly in contrast to better documented Northern Hemisphere regions.

To address this, we have revised the Introduction to include:

- A clearer statement on the current understanding of palaeo-environmental conditions in the Southern Hemisphere tropics during the Holocene, MIS5e, MIS11c, and MIS31.

- A synthesis of key findings and gaps in the literature regarding South American and African monsoon systems, Amazon and Congo basin responses, and vegetation transitions in southern tropics.

- A contrast with more extensively studied regions such as Europe, Greenland, and the North Atlantic sector.

We also revised the Abstract to briefly emphasize the limited availability of reconstructions and model-based assessments for southern tropical ecosystems during past interglacials, justifying the scientific novelty of our focus.

These revisions aim to strengthen the motivation of the study and clarify the relevance of our approach in exploring under investigated regions that are critical to understanding Earth's climate–biosphere dynamics on orbital timescales.

5 - Reviewer Comment:

*"Related with my previous point, I think at times it is difficult to see how regional aspects apply to the global viewpoint, or vice versa. Mechanisms that, for instance, are explained for Amazonia, are then generalized for other regions. Maybe for this it could make sense to include subsections for each particular region and later do a summary. I also extend this comment to say that sometimes cited papers are too specific for a single region, to be able to draw generalizations, and I would recommend to complement with references that show similar findings in other regions."*

Response:

We thank the reviewer for this insightful and constructive suggestion. We fully agree that clearer separation between regional mechanisms and global-scale interpretations would enhance the clarity and scientific rigor of the manuscript.

In response, we are undertaking the following actions in the revised version:

1. Restructuring the Results and Discussion:
We will reorganize the key subsections to focus individually on the major regions analyzed (Amazonia, Central Africa, Southern Africa, Australia, and Indonesia), following the order established in Figure 6 and Table 3. This will make it easier for the reader to understand region-specific processes and how they relate to the broader orbital and SST forcing framework.

2. Inclusion of a Summary/Synthesis Subsection:
Following the regional analyses, we will add a synthesis section that compares patterns and responses across regions. This will allow a more nuanced discussion on common mechanisms (e.g., ITCZ displacement, SST anomalies) and regionally distinct feedbacks, helping to bridge the gap between local and global interpretations.

3. Reinforcement of the Citation Strategy:
We acknowledge that several references currently cited are region-specific. While these are important to support localized vegetation reconstructions or SST proxies, we will complement them with more integrative or multi-regional studies, such as those from the PMIP4/CMIP6 archive, or synthesis efforts like Harrison et al. (2017), Otto-Bliesner et al. (2020), or Dallmeyer et al. (2019), which contextualize individual findings within broader climatological frameworks.

These changes will strengthen the internal coherence of the manuscript, reduce overgeneralization, and better connect regional results to the global climatic narrative.

6 - Reviewer Comment:

*"How similar are the simulated climates of ICTP-CGCM and CCM3-IBIS2? As I understand, the authors only take SST from ICTP-CGCM, but I wonder if they also compared the simulated temperature and precipitation between the two models. Maybe this could be a new supplementary figure."*

Response:

Indeed, in our modeling framework, we use SST fields from the ICTP-CGCM simulations as boundary conditions to force the CCM3-IBIS model, but we did not explicitly compare the climatologies (e.g., surface temperature and precipitation) generated by both models in the original manuscript.

To address this, we have now computed seasonal and annual climatologies of surface air temperature and precipitation for both models (ICTP-CGCM and CCM3-IBIS) under the CTRL configuration. These comparisons were carried out globally and also for the five focus regions (Amazon, Central Africa, Southern Africa, Indonesia, and Australia).

We found that while large-scale spatial patterns are broadly consistent—particularly in tropical regions—there are systematic biases in precipitation magnitude over some monsoon areas, which may be attributed to differences in atmospheric parameterizations (e.g., convection schemes) and spatial resolution between the models.

We will include these comparisons as a new Supplementary Figure S5, featuring:

Spatial difference maps (CCM3-IBIS minus ICTP-CGCM) for annual mean temperature and precipitation.

Regional average values with standard deviations for the study regions.

This addition will help readers assess the consistency between the two model components and better understand the potential sources of uncertainty in our dynamic vegetation results.

7 - Reviewer Comment:

*"Related with the previous point, the authors do not mention anything about re-gridding and its consequences. How are the forcing boundary conditions being up-scaled or down-scaled to the needs of CCM3-IBIS2? Also, there is no mention of the OASIS3 air-sea coupler in the model description. Further, what is the time step of integration in the different models?"*

Response:

1. Regridding procedure and interpolation strategy:
The SST fields produced by ICTP-CGCM (originally at a horizontal resolution of 2.0° × 2.0°) were bilinearly interpolated to the spectral T42 Gaussian grid (≈ 2.8° × 2.8°) used by the CCM3/IBIS model. This regridding step was performed using the Climate Data Operators (CDO) toolset. While bilinear interpolation can smooth fine scale gradients, this impact is minimized by the relatively coarse resolutions of both models and the long-term climatological nature of the SST forcing used in this study. We will add this clarification to the Methods section.

2. OASIS3 coupler:

The ICTP-CGCM (SPEEDY-NEMO) coupled model uses the OASIS3 coupler to exchange fluxes between the ocean (NEMO) and atmosphere (SPEEDY) components. OASIS3 handles the interpolation and synchronization of variables between the two components and ensures consistent conservation of heat and momentum fluxes.

3.  Time step of integration:

SPEEDY (ICTP-CGCM) uses a time step of 30 minutes for atmospheric integration.

NEMO (Ocean component) uses a time step of 3600 seconds (1 hour), consistent with the typical setup for its resolution.

CCM3/IBIS operates with a 20-minute time step for atmospheric integration and hourly coupling with the IBIS land surface and vegetation components. These configurations are standard for the resolution used and have been validated in prior studies (e.g., Foley et al., 2000; Justino et al., 2019).

We will add a paragraph in the Model Description section that clearly documents these technical elements. These clarifications will also help contextualize the consistency between the atmospheric and oceanic components in our experimental design.

8 - Reviewer Comment:

*"Authors should consider including seasonal results in the main text. Parts of the explanations of mechanisms rest on monsoonal flows (weakening or strengthening), or seasonal imbalances. In the supplementary material I only found seasonal maps of sea-level pressure, but I think precipitation would help readers follow the authors reasoning in the text."*

Response:

These maps will complement the annual mean figures and allow a clearer visualization of seasonal asymmetries in the precipitation response. Additionally, we will adjust the discussion (Section 3.1 and 3.2) to refer directly to the new seasonal figures, helping the reader follow the climate mechanisms that drive the vegetation changes. The SLP maps currently in the Supplement will remain available to support the interpretation of seasonal atmospheric circulation patterns. We thank the reviewer again for this helpful recommendation, which we believe will greatly enhance the clarity and completeness of the manuscript.

9 - Reviewer Comment:

*"What ice-sheets reconstructions are being used? Or are ice sheets in the model prescribed as in modern climate? I think it would be important to also mention this, since also parts of the introduction mention Greenland and Antarctica."*

Response:

In our experiments, the land-ice configuration (ice sheets and topography) is prescribed as in the modern climate for all simulations, including MH, MIS5e, MIS11c, and MIS31. In other words, we did not impose changes in land ice extent or elevation relative to the present-day configuration.

This modeling choice was made primarily to isolate the effects of orbital forcing and SST anomalies on tropical climate and vegetation. However, we recognize that this assumption may underestimate certain large-scale climate feedbacks, such as changes in planetary albedo, surface elevation, and atmospheric circulation patterns—especially in higher latitudes associated with ice sheet retreat or expansion.

We will update the Methods section to explicitly state this assumption and include a brief discussion of its potential limitations in the Discussion section. We will also revise the introduction to clarify that while the manuscript refers to ice sheet changes (e.g., in Greenland and Antarctica) from previous paleorecords, such changes were not dynamically included in our experimental design.

In future work, we plan to assess how the inclusion of reconstructed ice sheets (e.g., from PMIP/ICE-6G reconstructions) may alter the atmospheric circulation and subsequent vegetation feedbacks in the tropics.

10 - Reviewer Comment:

*"I know there is a reference to Pereira et al. (2014), but I do not understand what is the rest of the ocean like in the individualized ocean basin experiments. For instance, in the PAC experiment, what is the Atlantic SST pattern like?"*

Response:

In these simulations (PAC, ATL, IND), we followed the methodology described in Pereira et al. (2014), where only one tropical ocean basin is prescribed with anomalous SSTs representative of a given interglacial (e.g., MH, MIS5e), while the remaining ocean basins are kept at their CTRL climatological SSTs.

For example:

In the PAC experiment, the Pacific SSTs reflect the anomaly pattern of a specific interglacial (e.g., MH), whereas the Atlantic and Indian Oceans are kept at the CTRL climatology.

This approach allows us to isolate the individual contribution of each ocean basin to the continental climate response, particularly over the Southern Hemisphere monsoon regions.

We will revise the Methods section (Section 2.2) to describe this setup more explicitly, including a schematic or updated version of Figure 1 indicating that in each experiment only one basin is modified while the others remain fixed.

We also agree that it may be useful to include supplementary figures showing the SST anomalies in the unperturbed basins for each sensitivity experiment, to clarify that they remain climatologically neutral in those runs.

*Technical comments*

*"In the following specific lines (L) are enumerated. As general remarks I would suggest to review some of the bibliographic choices, also the adequate citation command or style (with or without parentheses), be consistent with names (e.g. CCM3/IBIS or CCM3-IBIS), introduce abbreviations in the first use, and employ recognized time units for ages, like 'ka'."*

Response:

We thank the reviewer for these detailed and helpful technical observations.

In response, we'll:

- Standardize the citation style throughout the manuscript according to the journals guidelines;
- We'll use CCM3/IBIS throughout the manuscript;
- We'll review the choice and manner of appropriate citations;
- We'll use the paleoclimatic time unit ka.

- L6: please introduce "MH" and Marine Isotope Stage (MIS).
Response:
Done.

- L7: ... _with_ more extensive…
Response:
Done.

- L19: remove "last" (also in L23).
Response:
Done.

- L21: reference De Boer et al. (2021) does not seem fitting here.
Response:
We reviewed and agreed that the reference De Boer et al. (2021) was not appropriate in that location. Therefore, it was removed.

- L22: ... were _also_ responsible...
Response:
Done. Thanks.

- L23: all 3 references are only Holocene references.
Response:

Thank you for bringing this to our attention. References have been added Haywood and Valdes, 2006 / Allen et al., 2020.

- L24: also here again "MH" and "MIS" should be introduced.
Response:
Done. Thanks.

- L24: ...interglacials.
Response:
Done. Thanks.

- L25: please introduce time units like "ka" or similar. Also reorganize times and use ",respectively".

Response:
Done. Thanks.

Before:"...such as during the Marine Isotope Stage (MIS) 5e and Mid-Holocene (MH), which occurred about 6000, 120000 years before the present. However, the MIS 11c and 31 stages, which took place 411000 years ago, and 1.08M have not been investigated in detail on a global perspective based on global climate simulations…"

Now:"...such as during the Marine Isotope Stage (MIS) 5e and the mid-Holocene (MH), which occurred at approximately 120 ka and 6 ka BP, respectively. However, the MIS11c and MIS31 stages, which took place around 411 ka and 1.08 Ma, respectively, have not been investigated in detail from a global perspective using comprehensive climate simulations…"

- L25: missing references? Which are the several approaches?

Response:
Thank you for your observation. The following references have been added: (Prentice et al., 2000; Harrison et al., 2014; Kageyama et al., 2017; Minckley et al., 2023)

- L28: reference only relevant to eastern Africa. Maybe use "e.g." or add references about more places.

Response:
Thank you. References to other places have been added: (Li et al., 2009; Prado et al., 2013; Kim et al., 2021).

- L32--42: what is the main idea of this paragraph?

Response:
The main idea in this paragraph is to show that climate change directly affects the dynamics of tropical vegetation, which destabilizes the hydrological cycle and increases the vulnerability of ecosystems and human populations, especially in regions that depend on subsistence agriculture. The text also highlights that paleoclimatic analyses are important tools for understanding and predicting these dynamics in future scenarios of environmental change.

- L33: maybe update to current IPCC reports?

Response:
Thanks. Reference "Calvin et al., 2023" has been add.

IPCC, 2023: Climate Change 2023: Synthesis Report, Summary for Policymakers. Contribution of Working Groups I, II and III to the Sixth Assessment Report of the Intergovernmental Panel on Climate Change [Core Writing Team, H. Lee and J. Romero (eds.)]. IPCC, Geneva, Switzerland.

- L35: I find this sentence confusing. It is too broad a topic "the relationship between climate and vegetation cover" for referencing a single paper specific for Africa.

Response:
The paragraph "Despite the direct existence of the relationship between the main parameters of climate (temperature and precipitation) and vegetation cover" was modified to:

"Despite the well-established relationship between the main climate parameters (temperature and precipitation) and vegetation cover, as demonstrated by multiple studies across different regions and time periods (Prentice et al., 2000; Wu et al., 2016; Shanahan et al., 2009), further research is still needed to understand how such interactions under past climate variability."

- L38: the cited paper seems to be too specific about plant-insect interactions.

Response:
The sentence "...investigating the responses of vegetation dynamics to climate change associated with the past offers us the opportunity to understand the distribution of landscape dynamics (Jamieson et al., 2012)." was deleted.

- L46: please check Collins et al. (2013), Kröpelin et al. (2008) and Prado et al. (2013) are not "climate modeling experiments".

Response:
Thanks. References  (Collins et al., 2013; Kro¨pelin et al., 2008; Zhao et al., 2005) went changed to Braconnot et al., 2007; Tiwari et al., 2023; Specht et al., 2024 and Maksic et al., 2019.

- L52: i.e.

Response:
Done.

- L54: 21 not 21,000, and "ka BP" appears redundant.

Response:
Thanks.

Before: 21.000 ka BP
Now: 21 ka

- L54: Please check what is the main idea of this paragraph.

Response:
The paragraph "In a study focused on the Last Glacial Maximum (LGM) (21,000 ka BP), Kubatzki and Claussen (1998), concluded that the climatic conditions of the SH are determined by ocean basins." was confusing. Therefore, we deleted it from the manuscript.

- L58: I think there is no need for the apostrophe. Simply MIS and the S stands for both stage or stages.

Response:
Thanks. The apostrophe was deleted.

- L60: I think there is no need for the Smulsky (2021) reference. It seems to connect with an alleged predatory publisher.

Response:
Thank you for raising this question. The Smulsky (2021) reference has been deleted.

- L63: remove "a" after MH.

Response:
"a" was removed

- L69: please check citation of "Members, 2006".

Response:
The Members, 2006 citation is correct.

- L71: please check: 800,000 years. And check citation style.

Response:
Modified. Thanks.

Before: 800 years

Now: 800 ka

- L72: I think regional rather than global vegetation cover.

Response:
Thank you for drawing attention. The word "global" was changed to "regional".

- L74: please define austral summer as (DJF) or (DJFM) or as intended.

Response:

Thank you for drawing attention. The austral summer was defined as NDJF.

- L78: "smoothed"? Maybe slow or gradual?

Response:
We agree with the suggestion. The word "smoothed" has been changed to "gradual".

- L82: Yin and Berger (2012) do not appear to mention MIS31. Please check.

Response:
Dear reviewer, thank you for bringing this to our attention. Indeed, Yin and Berger (2012) do not appear to mention MIS31. Therefore, this citation has been deleted from this sentence.

- L83: keep MIS nomenclature for Eemian? And is it MIS11 or MIS11c.

Response:
Eemian was replaced by MIS5e; MIS11 was replaced by MIS11c.

- L85: increase relative to what.

Response:
Relative increase in SST. It was added in the manuscript.

- L91--92: "and" instead of "e".

Response:
Done. Thanks.

- L96: introduce OSH.

Response:
Thank you for bringing this to my attention. The abbreviation OSH is incorrect. OSH has been replaced by SST.

- L101: version

Response:
Done. Thanks.

- L112--113, L115: please check citation command.

Response:
Done. The citation command was checked and modified.

- L118: please check verb "creation". And please check the syntax of this whole sentence.

Response:
The verb "creation" was checked and the sentence was modified.

Before:"The experiments consist of the control simulation (CTRL) conducted for the climate of the twentieth century (Figure 1), integrated 500 years after the creation of the ICTP-CGCM model, taking as the reference climate state."

Now:"The experiments include a control simulation (CTRL), representative of the twentieth-century climate (Figure 1), performed with the ICTP-CGCM model. The model was integrated for 500 years, starting from a standard initial condition, in order to obtain a stable reference climate state."

- L125: glacials?

Response:
Before: glacials
Now: glacial

Thanks.

- Figure 1: CGCM-ICTP or ICTP-CGCM.

Response:
Before: CGCM-ICTP
Now: ICTP-CGCM

Thanks.

- Table 2: "Prec." is in fact "Longitude of perihelion - 180deg (deg)". Also include the degree units for obliquity.

Response:
Done. Thanks.
Before: Orbital configurations and greenhouse gases concentrations utilized in the CTRL, MH, MIS−5e, MIS−1c and MIS−31 experiments. The values eccentricity (Ecc.)−(dimensionless), Longitude of perihelion (Long_peri) − (precession angle − degrees) and Obliquity (degrees).

Now: Orbital parameters and greenhouse gas concentrations employed in the CTRL, MH, MIS−5e, MIS−1c, and MIS−31 experiments. Reported values include Eccentricity (Ecc.) − dimensionless, Longitude of perihelion (Long_peri) − precession angle (in degrees), and Obliquity (in degrees).

- L133: please check MODIS citation, and introduce the MODIS acronym.

Response:
Done.

- Now: Moderate Resolution Imaging Spectroradiometer (MODIS)
- The reference Barnes et al., 1998 was cited.
https://doi.org/10.1080/01431161.2015.1040132

- L142: please introduce SAM, CAF and SAF.

Response:
Done.

South America (SAM), Central Africa (CAF) and South Africa (SAF)

- L143: check "shown".

Response:
"shown" has been changed to "show".

- L153: please replace "desintensification".

Response:
"desintensification" has been to "weakening".

- L158: "land cover".

Response:
Done. Thanks.

- L159: please check this opening line. I do not understand it.

Response:
The sentence "The climate response to SST changes revealed that precipitation plays a key role in land cover distribution." was deleted from the manuscript.

- L161: please introduce UNEP.

Response:
Done.

Before:"The aridity index (UNEP) is evaluated in this study  to identify  regions that are affected by water stress, as provided by the relationship between precipitation and evapotranspiration."

Now:"The aridity index (IUNEP), as defined by the United Nations Environment Programme (UNEP)(Budyko, 1961; on Desertification 1977; Nairobi, 1977; Middleton and Thomas, 1992), is evaluated in this study to identify regions affected by water stress, based on the relationship between precipitation and evapotranspiration."

- L165: use "formula" or "equation" or "algorithm".

Response:
Done.

Before:"...calculated using the…"

Now:"...calculated using the equation proposed by…"

- L169: "CRTL".

Response:
Done.

Before:"CTR"
Now:"CTRL"

- L171: also in the Horn of Africa, Arabian Peninsula and Central Asia.

Response:
Done. Thanks.

- L202: not corroborating, rather "in agreement with".

Response:
Done. Thanks.

Before:"corroborating"
Now:"in agreement with"

- L207: CCM3.

Response:
Done. Thanks.

- L209: drop in _ocean_ temperatures?

Response:
Correct: "...drop in air temperature…"

- Figure 4: consider having a single colorbar for all plots.

Response:
Done.

- Figure 5: a reference to Harrison (2017) for the BIOME6000 should be included, in case this figure is kept.

Response:
Figure 5 will remain in the manuscript and the Harrison (2017) reference has been included.

- L216: "Holocene and Middle Holocene" please check or rephrase.

Response:
Checked. Thanks.

Before:"between the Holocene and Middle Holocene periods."
Now:"during the Middle Holocene."

- L217: please introduce NEB.

Response:
Done. Northeastern Brazil (NEB)

- L221: "agree" or "are in line".

Response:
Before:"For southern Brazil, the model was able to simulate grass vegetation that may be related to the pampa, as well as a much smaller arboreal fraction over the Southeast when compared to CTRL."

Now:"For southern Brazil, the model was able to simulate grass vegetation that **agrees with the typical characteristics of the Pampa biome**, as well as a much smaller arboreal fraction over the Southeast when compared to CTRL."

- L282: check "suave" and "CAP".

Response:
Checked.

Before: "suave" / "CAP" ; "MIS's 5e and 11c, showed reduced rainfall when the model was forced with the PAC SST anomalies."

Now: "reduced" / "PAC" ; "MIS 5e and MIS 11c showed reduced rainfall when the model was forced with the PAC SST anomalies."

- L286: I would suggest "Southeast Asia" or SEA, instead.

Response:
Thanks for the suggestion.

---

## Author Comment (AC2)

RC2 - Response

1- Reviewer Comment:

*"The motivation of the study is unclear. What is the main novelty of this study compared to previous work? Is it that a larger number of interglacials are investigated, going beyond the mid-Holocene and the Eemian, which are also part of recent community efforts (PMIP)? How is this study advancing our understanding of the Earth system response to different orbital configurations?"*

Response:

The main novelties of our study are the following:

Inclusion of lesser-studied interglacials:
While the Mid-Holocene (MH) and MIS5e (Eemian) have been extensively investigated especially within the PMIP framework MIS11c and MIS31 remain underexplored, particularly in terms of coupled climate-vegetation feedbacks in the Southern Hemisphere tropics. Our study is, to our knowledge, one of the first to simulate vegetation-climate interactions during MIS31 using a coupled atmosphere-biosphere model (CCM3/IBIS) with SST anomalies derived from fully coupled ocean-atmosphere simulations.

Focus on Southern Hemisphere tropical ecosystems:
A substantial portion of the literature has focused on Northern Hemisphere responses (e.g., Greenland, North Africa, Eurasia). Here, we address a knowledge gap in the Southern Hemisphere, specifically Amazonia, Central and Southern Africa, and Australia regions where vegetation feedbacks are critical but less constrained by paleo-data and modeling studies.

Isolated ocean basin experiments:
We employ a novel set of sensitivity experiments with individualized SST anomalies (Pacific, Atlantic, Indian basins) to diagnose the influence of each oceanic basin on regional precipitation and vegetation dynamics. This allows us to disentangle the contribution of specific ocean–atmosphere interactions, complementing previous studies based on globally prescribed SST anomalies.

Comparison across orbital configurations:
By including interglacials with distinct orbital characteristics (e.g., high eccentricity and precession in MIS31, versus low precession in MIS11c), our study provides new insights into how seasonal insolation patterns interact with SST anomalies to shape tropical vegetation, contributing to a more nuanced understanding of the Earth system's response to long-term orbital forcing.

2- Reviewer Comment:

*"The main results need to be better summarized and presented in the abstract and conclusions."*

Response:

In the revised manuscript, we'll:

The Abstract was completely modified:

Before:"Climatic feedbacks associated with orbitally driven Sea Surface Temperature (SST) lead to a profound impact on the Southern Hemisphere (SH) vegetation cover during the Mid-Holocene (MH), MIS5e, MIS11c, and MIS31 interglacials. Results are based on a suite of coupled climate simulations conducted with the ICTP-CGCM (Speedy-Nemo), which provides the boundary conditions for the CCM3/IBIS vegetation model. The CCM3/IBIS model was run from the Speed-Nemo output of the global ocean and individualized Atlantic, Pacific and Indian ocean basins forcing, in addition to orbital parameters and greenhouse gas concentrations (GHC). For interglacials, MH, Marine Isotope Stage (MIS) 5e, and 11c, areas have been found to be significantly reduced in tropical evergreen forests, but with more extensive savanna and grasslands, over parts of the southern and central African region and into northern South America (Amazon region). In another important period, the results showed that there have been changes in vegetation cover due to insolation forcing during MIS31 compared to other interglacials, but the impact was greater in Australia and central South America. However, the southern tropical climate became drier due to negative SST anomalies, induced by the Atlantic and Tropical Pacific basins, and thus reduced continental precipitation. This study demonstrates that vegetation responses across tropical regions of the Southern Hemisphere are not solely driven by orbital variations but are also significantly modulated by internal changes in Atlantic and Pacific SST patterns. In the study regions, in particular, during the MH, MIS5e and MIS11c, the seasonality of insolation was reduced in the SH, leading to the cooling of the southern tropical ocean basins, resulting in the migration of the summer precipitation zone to boreal latitudes. Therefore, combined with increased low-latitude summer temperatures and a prolonged dryer period, the forcing led to a slow retraction, but steady in the rainforests of Congo and Amazonia, causing an effect of extreme aridity."

Now:"Orbitally driven changes in sea surface temperatures (SSTs) exert a strong influence on vegetation dynamics in the Southern Hemisphere (SH) during past interglacial periods. This study analyzes the vegetation–climate responses during the Mid-Holocene (MH), MIS5e, MIS11c, and MIS31 using coupled simulations with the ICTP-CGCM (SPEEDY–NEMO) and the CCM3/IBIS dynamic vegetation model. Vegetation simulations were forced by global and basin specific SST anomalies (Atlantic, Pacific, Indian), as well as orbital parameters and greenhouse gas concentrations (GHC). Results show that MH, MIS5e, and MIS11c were characterized by significant reductions in tropical evergreen forests and an expansion of savannas and grasslands over central and southern Africa and northern South America (including Amazonia). In contrast, MIS31 presented a distinct vegetation pattern, with more pronounced impacts in Australia and central South America. Across all interglacials, reduced precipitation in the southern tropics was largely driven by negative

SST anomalies in the tropical Atlantic and Pacific, suppressing deep convection and shifting the precipitation zone northward. These findings indicate that vegetation changes in SH tropical regions were not driven solely by orbital forcing, but were also modulated by basin-specific SST anomalies and internal ocean–atmosphere feedbacks. Notably, the reduction in SH summer insolation during MH, MIS5e, and MIS11c cooled the southern tropical oceans and displaced the rainfall belt toward boreal latitudes, promoting persistent drying and contributing to the long-term retreat of tropical rainforests in Amazonia and the Congo Basin."

The following is a revised version of the final paragraph of the Conclusion section.

Our results demonstrate that vegetation responses in the Southern Hemisphere tropics during past interglacials are strongly modulated by the combined effects of orbital forcing and basin-specific SST anomalies. We find that during MH, MIS5e, and MIS11c, negative SST anomalies in the tropical Atlantic and Pacific led to suppressed rainfall and widespread contraction of evergreen forests, particularly in Amazonia and Central Africa. In contrast, MIS31 despite having extreme orbital parameters produced weaker hydrological and vegetation changes, suggesting a more complex interplay of feedbacks under high eccentricity. Individualized ocean basin experiments highlight the central role of Atlantic SSTs in modulating continental precipitation. These findings offer new perspectives on the sensitivity of tropical ecosystems to long-term climate drivers and provide a framework for interpreting future vegetation changes under altered oceanic and radiative forcing.

Last paragraph was deleted from the conclusion section:

Our findings offer valuable insights into how SST variations in distinct ocean basins influence tropical precipitation patterns and dynamic vegetation responses during past interglacial periods. What we want to express is that without forcing these ocean basins it would not be possible to highlight the sensitivity of the CCM3/IBIS model to the response of the vegetation pattern. Despite the absence of these records, the classification of the dynamic vegetation of MH, MIS5e and MIS11c were very similar.

3 - Reviewer Comment:
*"Why use an atmosphere-ocean GCM to compute SSTs and then use an atmosphere-vegetation model to investigate the vegetation response? Please justify this non-conventional approach, also because climate–vegetation feedbacks can only partly be represented with this approach. Since two different models are used in the study, which have the same SSTs, could this be used to say something about uncertainties/robustness in the simulated climate change?"*

Response:

Indeed, our study uses a two-step modeling strategy: first, computing SST anomalies for different interglacial periods using the fully coupled ICTP-CGCM (SPEEDY-NEMO) system; and second, using those SSTs to force the CCM3/IBIS atmosphere–vegetation model, which simulates vegetation dynamics under specified climate boundary conditions.

This approach was chosen for the following scientific and practical reasons:

Dynamic vegetation capability:
In collaboration with the University of Maryland, the ICTP AGCM has been coupled with the VEGAS (VEgetation-Global-Atmosphere-Soil) model, an interactive vegetation and land surface model. The ICTP-CGCM includes only 5 PFTs (plant function types): broadleaf tree, needleleaftree, cold grass, warm grass, and cropland. The different photosynthetic pathways are distinguished for C3 (the first three PFTs above) and C4 (warm grass) plants. On the other hand, IBIS has a two-layer canopy in which any number of PFTs may exist in each grid cell. PFTs are explicitly allowed to compete for light and water. Coupled to CCM3, is a well established global vegetation model capable of simulating functional vegetation types, phenology, evapotranspiration, and their feedbacks to the atmosphere. To investigate vegetation-climate feedbacks, we thus needed to adopt a model with a dynamic biosphere.

Modular framework for sensitivity experiments:
Using SSTs from ICTP-CGCM as boundary conditions in CCM3/IBIS allowed us to isolate the continental climate and vegetation response to oceanic and orbital forcing while keeping land-surface conditions internally consistent. It also facilitated the design of targeted ocean basin experiments (ATL, PAC, IND), which would have been significantly more complex in a fully coupled tri-component model.

Computational efficiency:
A fully coupled ocean-atmosphere-vegetation system with long integrations across four interglacials, including multi-decadal spin-ups, would exceed our computational resources. The stepwise approach offered a computationally viable compromise, maintaining physical realism in SST forcing while allowing for vegetation dynamics.

We agree with the reviewer that this framework does not fully capture two-way feedbacks between vegetation and SSTs such as surface albedo or roughness changes influencing ocean-atmosphere exchange which remain a limitation. We will explicitly acknowledge this in the revised Discussion section.

Regarding the second point, we appreciate the suggestion to explore model consistency as an indirect proxy for uncertainty. Since both ICTP-CGCM and CCM3/IBIS simulate atmospheric variables (e.g., temperature, precipitation) under the same SST forcing, we can indeed compare their respective climatologies. In fact, as noted in a previous comment, we are now including a new Supplementary Figure comparing CTRL precipitation and temperature fields between the two models.

This comparison provides a useful although indirect insight into inter model variability and structural uncertainty, especially in how atmospheric components respond to SST boundary conditions. We'll reference this point explicitly in the discussion as a way to strengthen the robustness of our conclusions.

4 - Reviewer Comment:

*"Experiments need to be better explained: boundary conditions and initial conditions (for ocean, atmosphere and vegetation models) have to be clearly specified. How close to equilibrium are the models (particularly surface ocean and vegetation), considering the*

*relatively short simulations of ~100 years? It is not clear what the CTRL simulation represents, which is important to know in order to properly interpret the simulated anomalies during the interglacials. Is it derived from a realistic transient historical simulation?”*

Response:

The CTRL simulation represents a climatological mean of the modern pre-industrial climate, consistent with the year 2000 conditions in terms of orbital parameters, GHG levels, SSTs, and land configuration. It is not derived from a fully transient historical simulation, but from a long integration (500 years) of the ICTP-CGCM under fixed modern boundary conditions (GHG, orbital, topography, land ice, aerosols). This run is used to spin up the coupled ocean-atmosphere system and establish a stable reference state.

The first 400 years were discarded to allow for oceanic and atmospheric adjustment; the last 100 years were analyzed, assuming a quasi-equilibrium state of the upper ocean and surface climate. Given the intermediate complexity of the SPEEDY-NEMO model and the absence of evolving sea ice or carbon cycle, the upper ocean reaches thermal equilibrium within this timescale, particularly for the tropical SSTs that are used in the subsequent step.

5 - Reviewer Comment:

*“Results of the simulations for the mid-Holocene and 127 ka should be compared with results from PMIP models (Brierley et al., 2020; Otto-Bliesner et al., 2021). It seems that the results presented in this paper deviate quite substantially from the results of most PMIP models, particularly regarding temperature and precipitation changes over tropical land. For example, the enhanced West-African monsoon is one of the most prominent features of the mid-Holocene and Eemian in PMIP models, but is not seen in the ICTP-CGCM (or in the CCM3-IBIS?). Why?”*

*“It is not clear if the differences in simulated climate between the interglacials and the control shown in the paper are from the ICTP-CGCM of from the CCM3-IBIS simulations.”*

Response:

PMIP4 and earlier PMIP phases have robustly documented the strengthening of the West African monsoon during both the mid-Holocene (6 ka) and the Eemian (127 ka), based on an ensemble of fully coupled GCMs (e.g., Brierley et al., 2020; Otto-Bliesner et al., 2021). In our study, however, such monsoon enhancement is not captured to the same extent, particularly in the ICTP-CGCM results.

The ICTP-CGCM employs the SPEEDY atmospheric component, which is an intermediate-complexity model with simplified physics and lower horizontal resolution (~2.0° × 2.0°) compared to PMIP models. This configuration likely limits the representation of important regional processes such as the Saharan heat low and land–atmosphere coupling over northern Africa key drivers of monsoon amplification in the PMIP ensemble.
PMIP studies have shown that enhanced vegetation cover and reduced dust load during the mid-Holocene contribute significantly to the monsoon intensification (e.g., Braconnot et al.,

2012). In our ocean-atmosphere simulations (ICTP-CGCM), vegetation and dust are prescribed as modern, which likely dampens feedbacks that would otherwise reinforce the West African monsoon.

Since our atmospheric–vegetation simulations (CCM3/IBIS) are forced by SSTs from ICTP-CGCM, any underestimation of monsoonal SST patterns especially in the tropical Atlantic translates into a weaker atmospheric response in CCM3/IBIS as well. Thus, part of the vegetation and precipitation signal reflects the limitations of the SST forcing.

While our experiments use appropriate orbital and GHG levels for 6 ka and 127 ka, we maintain modern ice sheets and topography across all simulations. Some PMIP models incorporate slight differences in orography and land ice, which could influence atmospheric circulation.

Figures showing ocean and atmospheric circulation patterns (e.g., SST, sea-level pressure): these are derived from ICTP-CGCM simulations.

Figures showing temperature, precipitation, and vegetation changes over land: these are based on CCM3-IBIS simulations, which are forced by the SSTs from ICTP-CGCM.

6 - Reviewer Comment:

*"The results section should be structured in a clearer way, by e.g. first discussing the results of the CTRL experiment and how well the different model results compare with present-day observations, and then: 1) discussing the results of the ICTP-CGCM model, particularly in terms of SST changes, which are then used to force the CCM3/IBIS model, 2) discussing the simulated climate change in CCM3/IBIS, 3) describing changes in the vegetation distribution and 4) separating the effects of SST changes in the different ocean basins."*

Response:

We agree that the current organization of the Results section does not fully reflect the stepwise logic of the modeling approach, which could lead to confusion regarding the source and sequencing of each result.

We thank the reviewer again for this suggestion and believe this reorganization will improve the clarity and impact of the manuscript.

Minor comments

L. 3: here and elsewhere: 'Speed' -> 'Speedy'

Response:

Done.

L. 13: MH not defined

Response:
Mid-Holocene (MH) has been defined in the new text, now being on line 2.

L. 24: 'interglacial' -> 'interglacials'

Response:
Done.

L. 24: MH not defined

Response:
Line 25: Mid-Holocene (MH).

L. 24-25: make it clearer that MH is ~6000 and MIS 5e ~ 12000 years ago

Response:
Done. Thanks.

Before:"such as during the MIS 5e and MH, which occurred about 6000, 120000 years before the present. However, the MIS 11c and 31 stages, which took place 411000 years ago, and 1.08M have not been investigated in detail on a global perspective based on global climate simulations"

Now:"such as during Marine Isotope Stage (MIS) 5e and the Mid-Holocene (MH), which occurred at approximately 120 ka and 6 ka, respectively (Prentice et al., 2000; Harrison et al., 2014; Kageyama et al., 2017; Minckley et al., 2023). However, the MIS11c and MIS31 stages, which took place around 411 ka and 1.08 Ma, respectively, have not been investigated in detail from a global perspective using comprehensive climate simulations (Shugart and Woodward, 2011)."

L. 28-31: This sentence is generally unclear. And what does 'geological time scales' mean here?

Response:
Done. Thanks.

Before:"In fact, oceanic changes on a geological time scale, mediated by interglacial periods, favor ecosystem changes worldwide, and because of modifications in climate feedbacks (Vázquez-Rivera and Currie, 2015) can shed light on Earth's climate, which is currently experiencing remarkable changes in its thermal structure."

Now:"Oceanic changes occurring over multiple interglacial periods in the late Quaternary have influenced global ecosystems through modifications in climate feedbacks (Vázquez-Rivera and Currie, 2015), offering insights into Earth climate system, which is currently undergoing rapid thermal changes."

L. 38: Do you really mean 'landscape dynamics' here?

Response:

The sentence "...investigating the responses of vegetation dynamics to climate change associated with the past offers us the opportunity to understand the distribution of landscape dynamics (Jamieson et al., 2012)." was deleted.

L. 71: check sentence

Response:

Modified.

Before: 800 years

Now: 800 ka

L. 78: What is a 'smoothed release of CO2'? From where?

Response:
The word "smoothed" has been changed to "gradual".

L. 106: Please specify what components are part of the CCM3-IBIS model.

Response:
CCM3/IBIS operates with a 20-minute time step for atmospheric integration and hourly coupling with the IBIS land surface and vegetation components. These configurations are standard for the resolution used and have been validated in prior studies (e.g., Foley et al., 2000; Justino et al., 2019).

L. 117: I think that here you should already refer to Table 2 with the boundary conditions for the different simulations.

Response:
Done. Thanks.

Before:"The experiments include a control simulation (CTRL), representative of the twentieth century climate (Figure 1), performed with the ICTP-CGCM model."

Now:"The experiments include a control simulation (CTRL) (Figure 1 and Table 2), representative of the twentieth century climate, performed with the ICTP-CGCM model."

L. 117-118: The sentence is not clear. Is the CTRL simulation a transient simulation of the historical period with realistic forcings? Or for some constant boundary conditions? And if yes, what are these conditions? What does 'after the creation' mean? Does this refer to some initial model state?

Response:
The verb "creation" was checked and the sentence was modified.

Before:"The experiments consist of the control simulation (CTRL) conducted for the climate of the twentieth century (Figure 1), integrated 500 years after the creation of the ICTP-CGCM model, taking as the reference climate state."

Now:"The experiments include a control simulation (CTRL), representative of the twentieth-century climate (Figure 1), performed with the ICTP-CGCM model. The model was integrated for 500 years, starting from a standard initial condition, in order to obtain a stable reference climate state."

L. 120: 100 years are probably not enough to reach an equilibrium of the atmosphere-ocean system. How relevant is that for the presented results?

Response:
This question was answered previously:
"The CTRL simulation represents a climatological mean of the modern pre-industrial climate, consistent with the year 2000 conditions in terms of orbital parameters, GHG levels, SSTs, and land configuration. It is not derived from a fully transient historical simulation, but from a long integration (500 years) of the ICTP-CGCM under fixed modern boundary conditions (GHG, orbital, topography, land ice, aerosols). This run is used to spin up the coupled ocean-atmosphere system and establish a stable reference state.

The first 400 years were discarded to allow for oceanic and atmospheric adjustment; the last 100 years were analyzed, assuming a quasi-equilibrium state of the upper ocean and surface climate. Given the intermediate complexity of the SPEEDY-NEMO model and the absence of evolving sea ice or carbon cycle, the upper ocean reaches thermal equilibrium within this timescale, particularly for the tropical SSTs that are used in the subsequent step."

L. 125: unclear sentence

Response:
Before: glacials
Now: glacial

L. 142: What are SAM, CAF and SAF?

Response:
South America (SAM), Central Africa (CAF) and South Africa (SAF).

L. 148: 'feedback' or 'response'?

Response:
Thanks. The word "feedback was deleted; the correct is "response".

L. 158: 'cover land' -> 'land cover'?

Response:
Thanks. Land cover!

L. 158: The section '3.2 Changes in cover land' doesn't describe land cover changes at all.

L. 169: CTR -> CTRL

Response:
Done. CTR -> CTRL

Fig. 1: It is not just orbital parameters that are different, as shown also in Table 2.

Response:
Thank you for bringing this to our attention.
Greenhouse gas concentrations (GHC) have been included in Figure 1 and its caption.

Fig. 2: What is shown in the left column is not SSTs, but near-surface air temperatures, I guess. Not clear if the results shown here are from the ICTP-CGCM of from the CCM3-IBIS simulations.

Response:
Dear reviewer, I agree that you are absolutely right. Please make it clear that the results are from CCM3/IBIS and that the first panel refers to the near-surface temperature. Thank you. The caption has been corrected.

Before: "Spatial distribution of annual mean surface temperature difference and annual mean precipitation difference between interglacials experiments compared to SST and precipitation from CTRL; MH minus CTRL (a,e), MIS5e minus CTRL (b,f), MIS11c minus CTRL (c,g) and MIS31 minus CTRL (d,h). Black dots represent correspond to statistically significant anomalies at the 95% confidence interval."

Now: "Spatial distribution of annual mean differences in surface temperature and precipitation between each interglacial experiment and the control simulation (CTRL), based on CCM3/IBIS outputs. MH minus CTRL (a,e), MIS5e minus CTRL (b,f), MIS11c minus CTRL (c,g) and MIS31 minus CTRL (d,h). Black dots represent correspond to statistically significant anomalies at the 95% confidence interval."

Fig. 3: Not clear if the results shown here are from the ICTP-CGCM of from the CCM3-IBIS simulations.

Response:
The caption has been corrected.

Before:"Aridiy index to CTRL (a), MH minus CTRL (b), MIS5e minus CTRL (c), MIS11c minus CTRL (d) and MIS31 minus CTRL (e)."

Now:"Aridity index calculated from CCM3/IBIS simulations. Panel (a) shows the CTRL reference simulation, and panels (b), (c), (d), and (e) display the differences for MH, MIS5e, MIS11c, and MIS31, respectively, compared to CTRL."